# Generalizing Subgoals from Single Instances using Hypothesis-Preserving Ensembles

## Abstract

A reinforcement learning agent trained on a single source subgoal has no way to determine during training which features will be relevant for future instances of that same subgoal. This creates ambiguity: multiple plausible models of a subgoal can fit the training data but not all will successfully transfer. Humans address this ambiguity by maintaining alternative hypotheses until new information reveals the most effective one. Drawing inspiration from this, we introduce a hypothesis-preserving ensemble in which each member is a distinct, plausible subgoal classifier trained on the same source task. The agent then tests these alternative hypotheses in a new task, learning policies for the corresponding subtasks and uses task reward to select the most effective classifier. Experiments on Montezuma's Revenge and MiniGrid DoorMultiKey show that our method recovers subgoals learned in the source task, successfully adapting them to visually different tasks.

## 1 Introduction

Humans often solve new problems by identifying familiar structures: a repeated pattern, an intermediate milestone, a known bottleneck. Recognizing these elements allows us to adapt prior strategies instead of starting from scratch, a key driver of efficient generalization (Newell et al., 1972; Gick & Holyoak, 1983). Hierarchical reinforcement learning (HRL) (Barto & Mahadevan, 2003) offers a similar advantage: decomposing a task into subtasks, defined by subgoals, enables subtask recognition in new settings.

Most prior work on subgoal generalization assumes access to multiple training tasks or predefined goals (Hutsebaut-Buysse et al., 2022). We instead consider the case where all training data comes from a single task—a natural constraint in domains such as robotics and navigation (Nguyen-Tuong & Peters, 2011; Thrun, 2002). In this regime, multiple subgoal interpretations fit the training data equally well, yet only some transfer when environments change, an inherent ambiguity that cannot be resolved from limited data alone. Humans address this ambiguity by maintaining multiple competing hypotheses, discarding them only when new evidence appears (Anderson et al., 2015). Standard machine learning pipelines instead commit to a single model, risking overfitting to spurious correlations (Dietterich, 2000; De Haan et al., 2019). Rather than aggregating ensemble predictions to approximate a single correct answer, we preserve multiple distinct interpretations and defer commitment until future results discriminate among them—a fundamental paradigm shift from standard machine learning methods.

Our approach uses a hypothesis-preserving ensemble, where each member represents a distinct, plausible model of the target subgoal. A high-level policy tests these hypotheses in a new task, learning policies for the corresponding subtasks and using task reward as a guide for identifying the most effective classifier. This deferred commitment to a model improves generalization and enables subgoal reuse without retraining. We evaluate on Montezuma's Revenge and MiniGrid, showing that our approach improves subgoal generalization in new tasks, allowing for the reuse of task decompositions from prior experience. By explicitly representing and resolving subgoal ambiguity, we provide a principled framework for HRL that adapts discovered subgoals to new tasks without direct supervision.

## 2 BACKGROUND

We consider a Markov Decision Process $M = (S, A, r, p, \gamma)$, where $S$ denotes the state space, $A$ is the action space, $p(s_{t+1}|s_t, a_t)$ the transition dynamics and $r(s_t, a_t)$ a scalar reward function. The objective in RL is to find a policy $\pi(a|s)$ that maximizes the expected cumulative discounted reward. A task is an MDP $\mathcal{T} = (S, A, r_\mathcal{T}, p, \gamma)$ where $r_\mathcal{T}$ encodes the task goal $G_\mathcal{T}$. Complex tasks can be decomposed into subtasks, $t$, each defined by an intermediate subgoal $g_t$. A subgoal induces a reward function $r_g$, which returns 1 when the subgoal is satisfied and 0 otherwise. Task rewards are provided by the environment while subgoal rewards are derived from learned classifiers.

We focus on the regime where all training data is collected from a single training task. In this setting, the agent must learn a subgoal classifier from limited, homogeneous data that must hold in new tasks with different layouts or visual features. Robust subgoal generalization in this regime is necessary for the agent to recognize familiar subtasks in future tasks without retraining.

We build on hierarchical RL, where a high-level policy selects among temporally extended actions, **Options** (Sutton et al., 1999) in this work, as temporally extended actions defined by the tuple $(I_o, \pi_o, \beta_o)$. The initiation set, $I_o : S \rightarrow \{0, 1\}$, specifies where the option can start. The option policy $\pi_o : S \rightarrow A$ is a controller that transitions the agent from states in $I_o$ to states in $\beta_o$. The termination set, $\beta_o : S \rightarrow \{0, 1\}$ is the set of states in which option $o$ successfully terminates; a subgoal. The termination condition is a learned subgoal classifier. Generalizing this classifier beyond the training task allows for identifying familiar subtasks in new tasks.

## 3 RELATED WORK

**Subgoal Generalization and Recognition**   In symbolic AI, goal and plan recognition methods infer an agent's intent from partial observations (Kautz et al., 1986; Ramírez & Geffner, 2010; Baker et al., 2009), and hierarchical planners extend these ideas for richer temporal reasoning (Geib & Goldman, 2009). In RL, goal inference has been explored through universal value function approximators (Schaul et al., 2015a), hindsight experience replay (Andrychowicz et al., 2017) and unsupervised skill discovery (Eysenbach et al., 2018), as well as subgoal transfer in robotics (Kang & Kuo, 2025) and multi-agent settings (Xu et al., 2024). These approaches typically assume that data is available from multiple tasks, explicit goal conditioning or focus on generalization to changing reward functions. By contrast, we study generalization when all training data is drawn from a single task, requiring subgoal definitions to extend to unseen portions of the state space. Related HRL transfer methods such as portable options (Konidaris & Barto, 2007), successor features (Barreto et al., 2017) and the option keyboard (Barreto et al., 2019) also requires multiple tasks, assume known goal mappings or focus primarily on reward-function changes.

**Subgoal Discovery**   Early methods identify bottleneck states via diverse density (McGovern & Barto, 2001), betweenness centrality (Menache et al., 2002) or novelty measures (Şimşek & Barto, 2004). Later work leverages environment dynamics e.g. Laplacian option discovery (Machado et al., 2017) and successor-representation clustering (Jinnai et al., 2019). These focus on finding subgoals, not on generalizing them across environments from a single task.

**Generalization from Limited Experience**   Techniques such as auxiliary objectives (Jaderberg et al., 2016), contrastive representation learning (Laskin et al., 2020), domain randomization (Tobin et al., 2017) and meta-learning (Finn et al., 2017) improve robustness to distribution shifts. While effective for policy learning, they do not address the ambiguity of subgoal definitions when learning from limited data. Broader ML work on under-specification (D'Amour et al., 2022) formalizes this ambiguity.

**Ensemble Methods in RL**   Ensembles have been used for variance reduction (Wiering & Van Hasselt, 2008), exploration (Osband et al., 2016), model-based planning (Chua et al., 2018; Janner et al., 2019) and robustness (Lakshminarayanan et al., 2017; Lee et al., 2021). Our use differs in that we preserve multiple plausible subgoal definitions learned from a single task, deferring selection until new task data is available.

## 4 SUBGOAL GENERALIZATION WITH DATA FROM A SINGLE TRAINING TASK

We now consider the central question of this work: how can an agent, trained using data from only a single task, recognize the same subgoal in a new task which may differ in layout or appearance? This constraint models many real-world domains where collecting diverse training tasks is impractical, yet robust subgoal recognition can enable the reuse of learned skills. Limited and homogeneous data leaves the subgoal definition under-specified and this ambiguity must be resolved to generalize effectively.

In this regime, the agent observes one task $\mathcal{T}_{\text{train}}$ and a single instance of each discovered subgoal. Positive examples are states that satisfy the subgoal while negative samples are those that do not. Because all data comes from this single task, the learned definition must extend beyond the specific layout and appearance seen in training. At test time, the agent encounters a new task $\mathcal{T}_{\text{test}}$ that may differ substantially, and must decide whether states in this new task satisfy the same subgoal. Accurate recognition allows the agent to draw on prior information to improve performance without additional subgoal training.

### 4.1 UNDER-SPECIFICATION IN SUBGOAL LEARNING

When all training data comes from a single task, many features may consistently co-occur with a subgoal even if they are irrelevant. Without variation across tasks, the agent never encounters counter-examples that would separate spurious correlations from genuine defining features. This is an information-constrained problem: the data simply does not contain enough information to isolate the true subgoal definition, making it impossible for any one model—however sophisticated—to guarantee generalization.

Formally, let $\mathcal{S}$ be the state space, $\mathcal{C}$ a hypothesis class of binary classifiers $c : \mathcal{S} \rightarrow \{0, 1\}$ and $\mathcal{D}_{\text{subgoal}}$ a finite set of $N$ labeled states drawn i.i.d. from a single training-task distribution $P_{\text{train}}$ with $\text{supp}(P_{\text{train}}) \subsetneq \mathcal{S}$. We say $c^\star \in \mathcal{C}$ is identifiable from $\mathcal{D}_{\text{subgoal}}$ if

$$\forall c \in \mathcal{C}, \quad \left[ \forall (s, y) \in \mathcal{D}_{\text{subgoal}}, \ c(s) = y \right] \Rightarrow \ c = c^\star. \tag{1}$$

Since modern classifiers (e.g. deep nets) have VC dimensions $\gg N$, it follows that one can construct two functions in $\mathcal{C}$ that both fit all training points in $\mathcal{D}_{\text{subgoal}}$ but differ on at least one unseen state in $\mathcal{S} \setminus \mathcal{D}_{\text{subgoal}}$. Lemma 1 formalizes this non-identifiability (see Appendix A for proof).

**Lemma 1.** *Let $\mathcal{D}_{\text{subgoal}} = \{s_i\}_{i=1}^N$ and $\mathcal{U} = \mathcal{S} \setminus \mathcal{D}_{\text{subgoal}}$. If $\mathcal{U} \neq \emptyset$ and there exist $c_1, c_2 \in \mathcal{C}$ such that $c_1(s_i) = c_2(s_i) = y_i$ for all $(s_i, y_i) \in \mathcal{D}_{\text{subgoal}}$ but $c_1(u) \neq c_2(u)$ for some $u \in \mathcal{U}$, then no $c^\star \in \mathcal{C}$ is identifiable from $\mathcal{D}_{\text{subgoal}}$.*

This lemma makes precise that the version space $V(\mathcal{D}_{\text{subgoal}})$ contains multiple equally consistent subgoal definitions whenever any state lies outside the training set. A learner in this regime must commit to one of many plausible classifiers——exactly the ambiguity our hypothesis-preserving ensemble is designed to avoid.

Consider Figure 1: the left state satisfies a known subgoal, while the middle state does not. The agent must infer semantics of objects and their positions through environment interaction and observed reward. Now consider the state on the right: does it satisfy the subgoal? Several hypotheses are equally consistent with the training data—for example: (1) the presence of objects in the highlighted grid spaces (2) the presence of specific shapes (e.g. the circle) anywhere in the environment (3) the square is in a highlighted grid space. Each fits the training data but predicts differently for the unlabeled state.

This ambiguity is fundamental to this setting: any feature aligned with the subgoal during training will appear predictive, even if irrelevant elsewhere. Overcommitting to one hypothesis risks encoding task-specific features that fail to generalize. By maintaining multiple plausible hypotheses, we avoid prematurely discarding viable classifiers, resolving ambiguity later using information—such as task reward—from new tasks.

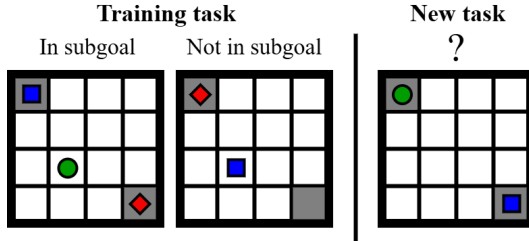

Figure 1: **Subgoal recognition from a single training task is inherently under-specified.** In the training task, the first state satisfies the subgoal while the second does not. Faced with the third state—drawn from a new task—the agent must decide whether it also satisfies the subgoal. With training data from a single task, multiple equally consistent definitions exist, and without resolving this ambiguity, subgoal classifiers may fail to generalize.

## 5 LEARNING GENERALIZING SUBGOAL CLASSIFIERS

Given a subgoal defined by data collected in a single training task, our goal is to determine whether that subgoal is satisfied in new tasks which may differ in layout or appearance. As shown in Section 4.1, this setting leaves the subgoal definition under-specified: multiple plausible interpretations fit the training data and no single model is guaranteed to be correct. To address this, we maintain a set of competing hypotheses, each representing a distinct, consistent definition of the subgoal's features. Preserving these alternatives reduces the risk of overfitting to spurious correlations and increases the chance that at least one hypothesis will transfer. Detecting previously identified subgoals in new tasks allows for reusing previously discovered structures without additional training.

Rather than committing to one classifier during training, we maintain an ensemble, deferring selecting until task-level information is available. We show

$$\mathbb{E}_T\left[\max_{c\in\mathcal{C}} R_T(c)\right] \ \geq \ \max_{c\in\mathcal{C}} \mathbb{E}_T\left[R_T(c)\right] \tag{2}$$

where $R_T(c)$ is the cumulative reward earned by running classifier $c$ in task $T$. Per-task hypothesis selection therefore can never underperform a fixed classifier (see Appendix A for more details).

### 5.1 HYPOTHESIS GENERATION

We model each subgoal as a binary classifier mapping states to 1 when the subgoal is satisfied and 0 otherwise. In this regime, the training data supports multiple plausible classifiers—each consistent with the observed examples but relying on different features. To preserve this ambiguity, we maintain a set of hypotheses, modeled as an ensemble of classifiers $\mathcal{C} = \{c_1, c_2, \ldots, c_k\}$ where each $c_n$ represents a distinct interpretation of the subgoal.

To encourage broader coverage, we promote diversity through one of two mechanisms. First, implicit diversity arises from random initialization. Second, we apply an explicit diversity objective using the DivDis algorithm (Lee et al., 2022), which encourages classifiers to disagree on unlabeled data by minimizing mutual information between classifiers:

$$L_{\mathrm{MI}}(c_i, c_j) \ = \ \sum_{y_i}\sum_{y_j} p_{ij}(y_i, y_j) \log \frac{p_{ij}(y_i, y_j)}{p_i(y_i)\,p_j(y_j)} \tag{3}$$

while maintaining low cross-entropy loss on labeled examples:

$$L_{\mathrm{xent}}(c_i) = \mathbb{E}_{x,y\in D_i}[\ell(c_i(x), y)]. \tag{4}$$

The mutual information term is computed over an unlabeled dataset $\mathcal{D}_{\mathrm{unlab}}$, gathered through exploration in the environment. Although this data lacks subgoal labels, they provide variation that helps classifiers develop complementary models and encourages diversity. Other diversity-promoting ensemble methods (e.g., D-BAT (Pagliardini et al., 2022)) could be used in place of DivDis, as our framework is agnostic to the specific ensemble learning technique

---

**Algorithm 1** Learning transferable subgoals and hypothesis selection via reward maximization

---

**Input:** $\mathcal{D}_{G_s}$, $\mathcal{D}_{\text{unlab}}$, max steps $\text{step}_{\max}$, option timeout $T_o$

  **Classifier training**
  randomly initialize all $f_i \in C$
  **for each** classifier $f_i \in C$ **do**
    train $f_i$ minimizing loss $L_{\text{xent}}$ on $\mathcal{D}_{G_s}$
    *For DivDis variant add additional loss term $L_{MI}$ on $\mathcal{D}_{unlab}$*
  **end for**
  **Policy initialization**
  Initialize $\pi_{o_i}$ for each $f_i \in C$, Initialize $\pi_h$
  **Policy training**
  $\text{step} \leftarrow 0$
  **while** $\text{step} < \text{step}_{\max}$ **do**
    $i \leftarrow \pi_h(s)$                                         $\triangleright$ $\pi_h$ selects option index
    $\pi_{\text{current}} \leftarrow \pi_{o_i}$, $t \leftarrow 0$
    **while** $f_i(s) \neq 1$ **and** $t < T_o$ **do**       $\triangleright$ execute until subgoal reached or timeout
      $a \leftarrow \pi_i(s)$                       $\triangleright$ get action from option policy
      $s \leftarrow$ execute $a$ in environment and observe $s'$
      $\text{steps} \leftarrow \text{steps} + 1$, $t \leftarrow t + 1$
      Update $\pi_{o_i}$ using subgoal pseudo-reward
    **end while**
    Update $\pi_h$ using task reward
  **end while**

---

This ensemble serves as a structured representation of the ambiguity inherent when training data is limited and homogeneous. Each classifier encodes a distinct, consistent interpretation of the subgoal, allowing the agent to defer commitment until task-relevant information reveals which definition generalizes best.

## 5.2 Reward-Guided Hypothesis Selection

Each hypothesis $c_n \in \mathcal{C}$ defines a distinct subgoal and we learn a corresponding option $o_n$ with policy $\pi_{o_n}$ to achieve the corresponding subtask. Each classifier $c_n$ induces a sparse reward function and $\pi_{o_n}$ is trained to maximize that reward, learning to achieve the hypothesized subgoal $c_n$.

Because the environment has no direct subgoal supervision—and the task reward, the only available signal of success, is not guaranteed to align with the learned subgoals—direct evaluation of which subgoal hypothesis is most appropriate is not possible. Instead, we use task reward as an indirect signal to assess which subgoal transfers most effectively. We define the most generalizable subgoal as the one whose associated option yields the highest cumulative reward in the current task—not because it is universally correct, but because it best aligns with the demands of the new task.

A high-level policy selects among the option policies to maximize task reward. By learning a Q-function over the option set, the agent implicitly identifies which subgoal hypothesis best supports task completion. We train both the high-level and option policies jointly (see Algorithm 1).

## 6 Experiments

Our experiments are designed to evaluate whether preserving multiple plausible subgoal hypotheses improves an agent's ability to recognize subgoal states under visual changes and effectively guide option policies. Specifically, we aim to answer:

1. **Data Efficiency:** How much labeled data is required for a subgoal classifier to correctly identify the same subgoal across visually distinct tasks?

2. **Hypothesis-Driven policy Learning:** Once a subgoal can be identified in a new task, can the agent learn an effective option policy for it?

3. **Task-Level Performance:** Does hypothesis-guided option learning improve the agent's ability to solve sparse-reward tasks compared to single-model baselines?

4. **Reward-Guided Disambiguation:** Can task reward reliably select the subgoal hypothesis that best matches the demands of the current task?

Each experiment isolates one of these questions, progressively building towards a fully integrated hierarchical agent that uses a hypothesis-preserving ensemble to guide policy learning for subgoals detected in new tasks.

We use two visually rich domains with pixel-based state spaces. Montezuma's Revenge (Bellemare et al., 2013; Machado et al., 2018) is used to test hypothesis quality in isolation, focusing on whether at least one preserved hypothesis correctly identifies the target subgoal in visually distinct tasks with different layouts. Minigrid DoorMultiKey (Chevalier-Boisvert et al., 2023) evaluates the full pipeline from subgoal recognition to option execution in a sparse-reward setting, reusing the same task decomposition as in training. To ensure controlled evaluation, all experiments use predefined subgoals, which both isolates subgoal recognition and hypothesis selection from the separate challenge of subgoal discovery, and enables direct comparison to a known ground-truth definition. See Appendix C for hyperparameters and experiment setup and pseudocode.

## 6.1 DATA EFFICIENCY

This experiment measures how the amount of labeled subgoal data affects the ability of our hypothesis-preserving ensemble to correctly identify a target subgoal across visually varied tasks. We isolate recognition performance from downstream control, focusing solely on whether at least one ensemble member generalizes beyond the training task.

Montezuma's Revenge is a visually complex Atari game made up of multiple rooms, each with distinct objects and layouts, making it an ideal domain for validating subgoal recognition. We define a `ClimbDownLadder` subgoal, which is satisfied when the agent is positioned at the base of a ladder. We incrementally expand the training set by adding labeled examples from additional rooms containing ladders. After each addition, all models are retrained and evaluated on data from all ladder rooms, including those not yet represented in the training set (see Algorithm 2 in Appendix C). Data from unseen rooms, with and without ladders, is used as unlabeled data. This experiment measures how increasing intra-task variation in the data affects generalization to unseen tasks.

We compare a single convolutional classifier with two hypothesis-preserving ensemble variants: a standard ensemble, which gains diversity through random initialization, and DivDis which encourages diversity explicitly during training. For ensembles, we report the accuracy of the best-performing member, reflecting the goal of retaining at least one valid hypothesis.

As shown in Figure 2, both ensembles outperform the single classifier when trained on data from a single room—the setting with highest ambiguity—indicating that maintaining multiple hypotheses increases the likelihood of capturing generalizing features. Note the CNN—a single model—barely outperforms random guessing on the binary classification problem. Accuracy improves sharply when labeled data from a second room is included, showing the benefit of even small increases in visual diversity. As more varied data is introduced, all methods show the same performance, validating the earlier theoretical result that deferring selecting is equal to or greater than learning a single model. The standard and DivDis ensembles achieve similar mean accuracy, with DivDis showing a slightly lower variance across seeds. These results support the claim that preserving multiple plausible hypotheses enables data-efficient subgoal identification in new tasks.

## 6.2 HYPOTHESIS-DRIVEN POLICY LEARNING

We next evaluate whether subgoal hypotheses can support the learning of effective option policies in new tasks. This step bridges subgoal identification and downstream control, testing whether a hypothesis learned from only the training task is accurate enough to serve as an effective termination condition when training a new policy from scratch in a visually different task.

We again use the Montezuma's Revenge `ClimbDownLadder` subgoal. As in the previous experiment, we incrementally add training data from each ladder room. After each addition, we train an option policy in each ladder room for the best performing ensemble member, using only the classifier

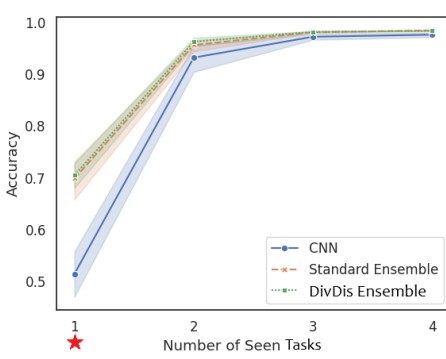

Figure 2: Accuracy of the best performing ensemble member as more labeled data is provided. Results are averaged over 10 seeds and bands represent standard deviation. Star indicates when only one task is provided during training.

Figure 3: Average Manhattan distance between policy termination point and the ground-truth subgoal; bars represent standard deviation over the last 100 option executions (lower is better) averaged over 10 seeds. Star indicates when only one task is provided during training.

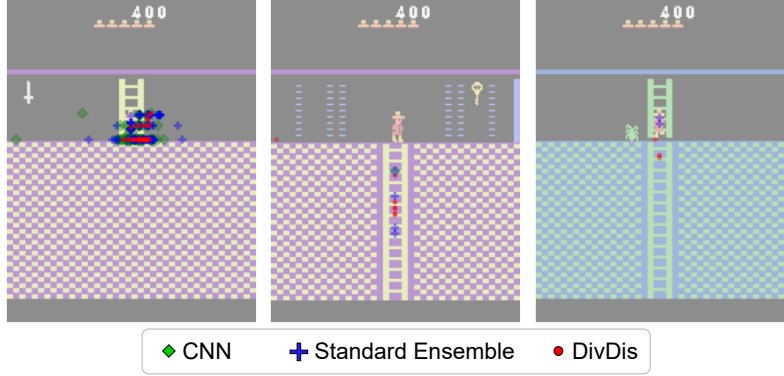

Figure 4: Scatter plot of termination locations for 100 skill executions of a single subgoal and policy, trained on data from two ladders for an unseen task. The agent begins atop the ladder in the middle room; climbing down leads to the right room, while moving left places the agent in the left room. Only the best-performing ensemble member is shown.

as the option termination condition. We use Deep Q-Networks (Mnih et al., 2015) (DQN) with prioritized experience replay (Schaul et al., 2015b) for the option policies, trained using a sparse reward defined by the learned classifier (1 in subgoal states and 0 otherwise), without ground-truth subgoal data. Each policy is trained for 300000 steps in each room which is sufficient for convergence.

Performance is measured by the average Manhattan distance between the policy termination state and the true subgoal location, averaged over 100 executions after policy training completes. We show the performance of each method, averaged over all ladder rooms, in Figure 3. Because this metric does not provide an intuitive idea of how useful the learned policies are, we focus on relative performance and provide a scatter plot (Figure 4) which shows where each policy terminated across the 100 evaluation executions, providing a qualitative visual analysis of classifier quality.

From Figure 3, both the standard and DivDis ensembles produce option policies whose terminations are closer to the true subgoal than those trained with a single CNN. When trained on data from a single task—which is our primary focus—both ensembles outperforming the CNN, with DivDis almost halving the average Manhattan distance achieved by the CNN. While the classifier accuracies

in Figure 2 showed little difference between the standard and DivDis ensembles, we see a clear performance gain from using explicit diversity for one room of training data during policy learning. The first room in Montezuma's Revenge has the largest visual difference to all other ladder rooms and thus shares the fewest features with the other tasks; in such cases, where visual differences are substantial, explicit diversity has a measurable effect. This effect diminishes as the training data better captures the variation present in future tasks.

Figure 4 shows the termination locations for an unseen task, for option policies trained using classifiers learned from two rooms of labeled data. The agent begins at the top of the ladder in the middle image and can either move left to reach a ladder base or climb down the ladder to the base shown in the right image. Only fully terminating ladders are provided during training, so the agent has never seen labeled examples resembling the right room, where the ladder continues through the floor. Nevertheless, both ensemble methods generalize to this variant, which is a valid `ClimbDownLadder` subgoal despite never being observed during training. By contrast, the CNN-based policy never terminates at this ladder base, failing to generalize to this case. While all classifiers occasionally misclassify termination states in the center of the ladder in the center figure, from the left image we see that the CNN is the most inconsistent—sometimes terminating when the agent is not close to the ladder—whereas both ensemble methods terminate consistently near the base.

These results validate the effectiveness of hypothesis-driven policy learning, reinforcing the claim that maintaining multiple hypotheses enables better subgoal generalization which can be leveraged for future policy learning. They also show that encouraging diversity during training improves subgoal detection, particularly when the training and test tasks differ substantially.

## 6.3 Task-Level Performance

Having demonstrated that hypothesis-driven subgoal generalization can support learning effective option policies, we now evaluate whether this task decomposition can be adapted to a new task. We use the Minigrid DoorMultiKey environment, a modification on the sparse-reward DoorKey task, where the agent must collect a key to unlock a door to reach the goal location, with additional distractor keys. This forces the hierarchical agent to distinguish between relevant and irrelevant subgoals as well as allowing for additional visual variation among tasks.

We define five subgoals for this task: `CollectBlueKey`, `CollectGreenKey`, `CollectRedKey`, `OpenRedDoor` and `GoToGoal`. Two of these subgoals— `CollectBlueKey` and `CollectGreenKey`—are not required to complete the test tasks. Including non-essential subgoals increases the decision complexity for the high-level policy, which must learn not only to select the most useful subgoal hypotheses but also to disregard subgoals that are irrelevant to the current task. Labeled data for all subgoals is collected from a single training task (seed 0), while unlabeled data is gathered from two additional seeds that are *not* included in the test task set.

Our hierarchical agent is trained as described in Algorithm 1. The high-level policy is a PPO agent (Schulman et al., 2017), that selects among option policies, each implemented as described in the previous option policy experiment. The action space consists of three hypotheses per subgoal; 15 available actions for the PPO agent.

We evaluate hierarchical agents using standard and DivDis ensembles, as well as a CNN-based option agent (five actions, one per subgoal). We no longer use only the best performing ensemble member and the hierarchical agent must determine which hypothesis best aligns with the current task. For reference, we include a hierarchical agent with oracle termination classifiers, representing the best achievable performance for the option-based agents. We also ablate the hierarchy by training flat DQN and PPO agents with access to only the primitive actions.

Figure 5 shows the average undiscounted episode reward. The PPO and DQN agents fail to complete the task with distractor keys which substantially enlarge the state space and make exploration difficult. The CNN-based option agent under-performs both ensemble-based methods, confirming that maintaining multiple hypotheses improves subgoal generalization. Both ensemble methods achieve near-optimal performance over time, closely matching the perfect-termination baseline. This confirms that we can reuse a previously beneficial task decomposition in new tasks by learning multiple hypotheses, selecting the best fitting hypothesis at test time.

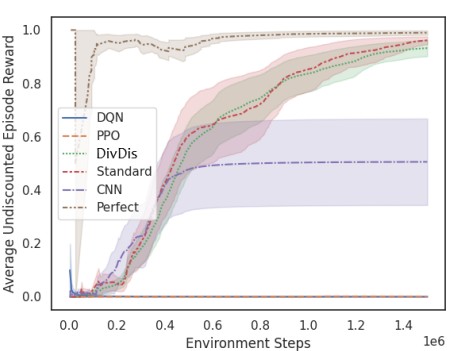

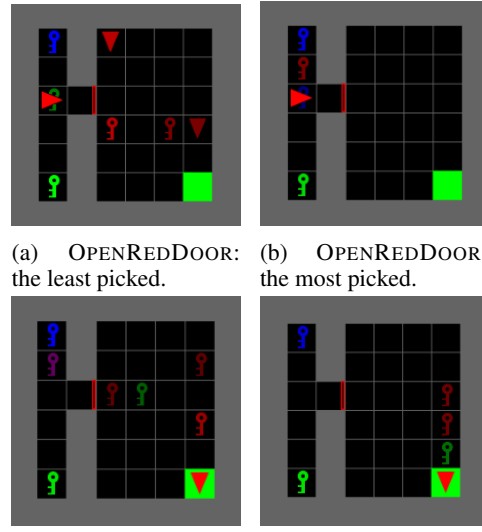

(a) OPENREDDOOR: the least picked.

(b) OPENREDDOOR: the most picked.

(c) GOTOGOAL: tied for most picked.

(d) GOTOGOAL: tied for most picked.

Figure 5: Average undiscounted reward for the modified MINIGRID DOORMULTIKEY environment. All results are averaged over 10 seeds and bands represent standard error. DivDis, Standard, CNN and Perfect are all option agents using the corresponding method for option termination classifiers. The perfect termination agent is the best performance we can expect from any option agent.

Figure 6: Overlaid termination states identified by members of the DivDis ensemble.

## 6.4 REWARD-GUIDED DISAMBIGUATION

To assess whether reward maximization can reliably identify the most useful subgoal hypothesis, we compare the termination sets of the most- and least-selected ensemble members in the MiniGrid DoorMultiKey environment. Figures 6a and 6b illustrate two members of the `OpenRedDoor` subgoal. The least-chosen hypothesis (Figure 6a) produces termination points scattered throughout the room, failing to consistently position the agent near the door. By contrast, the most-frequently selected hypothesis (Figure 6b) always terminates directly in front of the open door, closely matching the true subgoal. A similar pattern emerges for the `GoToGoal` subgoal in Figures 6c and 6d: highly selected hypotheses terminate exclusively at the goal position, whereas the least-selected ensemble member fails to identify any valid subgoal state in the new task and consequently never terminates its option policy successfully.

Across multiple subgoals, the high-level policy consistently favors hypotheses that lead to higher cumulative reward. This behavior shows that reward-driven selection acts as an implicit supervision signal, filtering out ineffective subgoal classifiers and retaining only those that support successful task completion. This mechanism allows the agent to defer commitment during training, then resolve subgoal ambiguity by selecting the hypothesis most aligned with the demands of the current task without requiring any subgoal labels in the target task.

## 7 CONCLUSION

We studied the problem of subgoal generalization when all available training data is drawn from a single task, where limited and homogeneous samples result in under-specified subgoals. We formalized this ambiguity and introduced a hypothesis-preserving ensemble that maintains multiple plausible hypotheses of a subgoal's defining features, deferring commitment until task-level evidence is available. Across Montezuma's Revenge and MiniGrid DoorMultiKey, this approach improves subgoal recognition, supports effective option learning without direct subgoal supervision and requires only task-reward to identify the most effective hypothesis. By explicitly representing and resolving ambiguity, our method provides a principled framework for adapting learned decompositions to new tasks under severe data constraints.

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

# A  PROOFS FOR IDENTIFIABILITY AND SUBGOAL GENERALIZATION RESULTS

## A.1  NON-IDENTIFIABILITY FROM DATA FROM A SINGLE TASK

Let $\mathcal{S}$ be a state space, $P_{\text{train}}$ a distribution on $\mathcal{S}$ with $\text{supp}(P_{\text{train}}) \subsetneq \mathcal{S}$ and $\mathcal{C} \subseteq \{c : \mathcal{S} \to \{0,1\}\}$ a hypothesis class. Assume realizability on observed data: labels are generated by some $c^\star \in \mathcal{C}$ so $y = c^\star$ for $s \sim P_{\text{train}}$. Assume there exist $c_1, c_2 \in \mathcal{C}$ such that

$$\forall s \in \text{supp}(P_{\text{train}}),\ c_1(s) = c_2(s), \quad \text{and} \quad \exists u \in \mathcal{S} \quad \text{such that} \quad c_1(u) \neq c_2(u).$$

Then for any finite $N \in \mathbb{N}$, with probability 1 over $\mathcal{D} = \{(s_i, y_i)\}_{i=1}^N \sim P_{\text{train}} \times \delta_{c^\star}$, the identifiability condition

$$\forall c \in \mathcal{C}, \quad \left[\forall (s_i, y_i) \in \mathcal{D},\ c(s_i) = y_i\right] \Rightarrow c = c^\star$$

fails. Consequently, $c^\star$ is not identifiable from data from a single task supported on $\text{supp}(P_{\text{train}})$.

*Proof.* With probability 1, all sampled states lie in the support: $\{s_i\}_{i=1}^N \subseteq \text{supp}(P_{\text{train}})$. On this event, for every $i$, $c_1(s_i) = c_2(s_i) = c^\star(s_i) = y_i$, so both $c_1$ and $c_2$ are consistent with $\mathcal{D}$. Since $c_1 \neq c_2$ on $\mathcal{S}$, at least one of them, we will call $c'$, differs from $c^\star$ somewhere in $\mathcal{S}$. Thus

$$\forall (s_i, y_i) \in \mathcal{D},\ c'(s_i) = y_i \quad \text{but} \quad c' \neq c^\star,$$

which violates the identifiability condition. Because the event holds with probability 1, identifiability fails almost surely for any finite N. $\square$

## A.2 Lemma 1: Subgoal Under-Specification

Let $\mathcal{D}_{\text{subgoal}} = \{s_i\}_{i=1}^N$ and $\mathcal{U} = \mathcal{S} \setminus \mathcal{D}_{\text{subgoal}}$. If $\mathcal{U} \neq \emptyset$ and there exist $c_1, c_2 \in \mathcal{C}$ such that $c_1(s_i) = c_2(s_i) = y_i$ for all $(s_i, y_i) \in \mathcal{D}_{\text{subgoal}}$ but $c_1(u) \neq c_2(u)$ for some $u \in \mathcal{U}$, then no $c^\star \in \mathcal{C}$ is identifiable from $\mathcal{D}_{\text{subgoal}}$.

*Proof.* Assume there exists some $c^\star$ which is identifiable from $\mathcal{D}_{\text{subgoal}}$, then by definition:

$$\forall c \in \mathcal{C}, \quad \big(c \in V(\mathcal{D}_{\text{subgoal}})\big) \implies c = c^\star.$$

This means $V(\mathcal{D}_{\text{subgoal}}) = \{c^\star\}$, i.e. $|V(\mathcal{D})| = 1$. Recall that the version space $V(\mathcal{D}_{\text{subgoal}}) = \{c \in \mathcal{C} \mid \forall(s_i, y_i) \in \mathcal{D}_{\text{subgoal}}, c(s_i) = y_i\}$.

Assume there exists two distinct $c_1, c_2 \in \mathcal{C}$ both agreeing on every $(s_i, y_i) \in \mathcal{D}_{\text{subgoal}}$. So $c_1, c_2 \in V(\mathcal{D})$ and $c_1 \neq c_2$, so $|V(\mathcal{D}_{\text{subgoal}})| \geq 2$.

**Contradiction:** These two assumptions contradict each other. Therefore no $c^\star \in \mathcal{C}$ is identifiable from $\mathcal{D}_{\text{subgoal}}$.

$\square$

## A.3 Deferred Selection

Let $\mathcal{C} = \{c_1, \ldots, c_K\}$ be our ensemble of subgoal classifiers, and let $R_T(c)$ be the cumulative reward obtained by running classifier $c$ on task $T$. Then

$$\mathbb{E}_T\Big[\max_{c \in \mathcal{C}} R_T(c)\Big] \geq \max_{c \in \mathcal{C}} \mathbb{E}_T\big[R_T(c)\big].$$

Equality holds if and only if there is a single hypothesis $c^\star \in \mathcal{C}$ that maximizes $R_T(c)$ for almost every task $T$. In that case, per-task selection reduces to always choosing $c^\star$.

*Proof.* Define the random vector $X = (X_1, \ldots, X_K)$ by $X_i = R_T(c_i)$ with $T \sim \mathcal{T}$.

For each coordinate i

$$X_i \leq \max_{1 \leq j \leq K} X_j \quad \implies \quad \mathbb{E}[X_i] \leq \mathbb{E}[\max_j X_j].$$

Taking the maximum over i yields $\max_i \mathbb{E}[X_i] \leq \mathbb{E}[\max_j X_j]$, which is exactly

$$\max_{c \in \mathcal{C}} \mathbb{E}_T[R_T(c)] \leq \mathbb{E}_T\big[\max_{c \in \mathcal{C}} R_T(c)\big].$$

$\square$

# B Ensemble Head Saliency Results

# C Experiment setup

## C.1 Experiment pseudocode

## C.2 Compute resources

All MONTEZUMASREVENGE experiments were run using 1 Nvidia GTX 4090 GPU and 1 AMD Ryzen Threadripper PRO 5995WX 64-Cores cpu each, for a total of 64 cores. Each run used 126GB RAM. A single run takes under 12 hours to run.

All MINIGRID DOORMULTIKEY experiments were run with 2 Nvidia GTX 4090 GPUs and 2 AMD Ryzen Threadriper PRO 5995WX 64-Cores (128 cores) per run. Each run used 252GB RAM. A single run with an ensemble takes around 24 hours to complete.

All experiments were run on a 10 node cluster, each node has 2 Nvidia GTX 4090 GPUs and 2 AMD Ryzen Threadriper PRO 5995WX 64-Cores CPUs. All computers run Ubuntu 22.04.3 LTS.

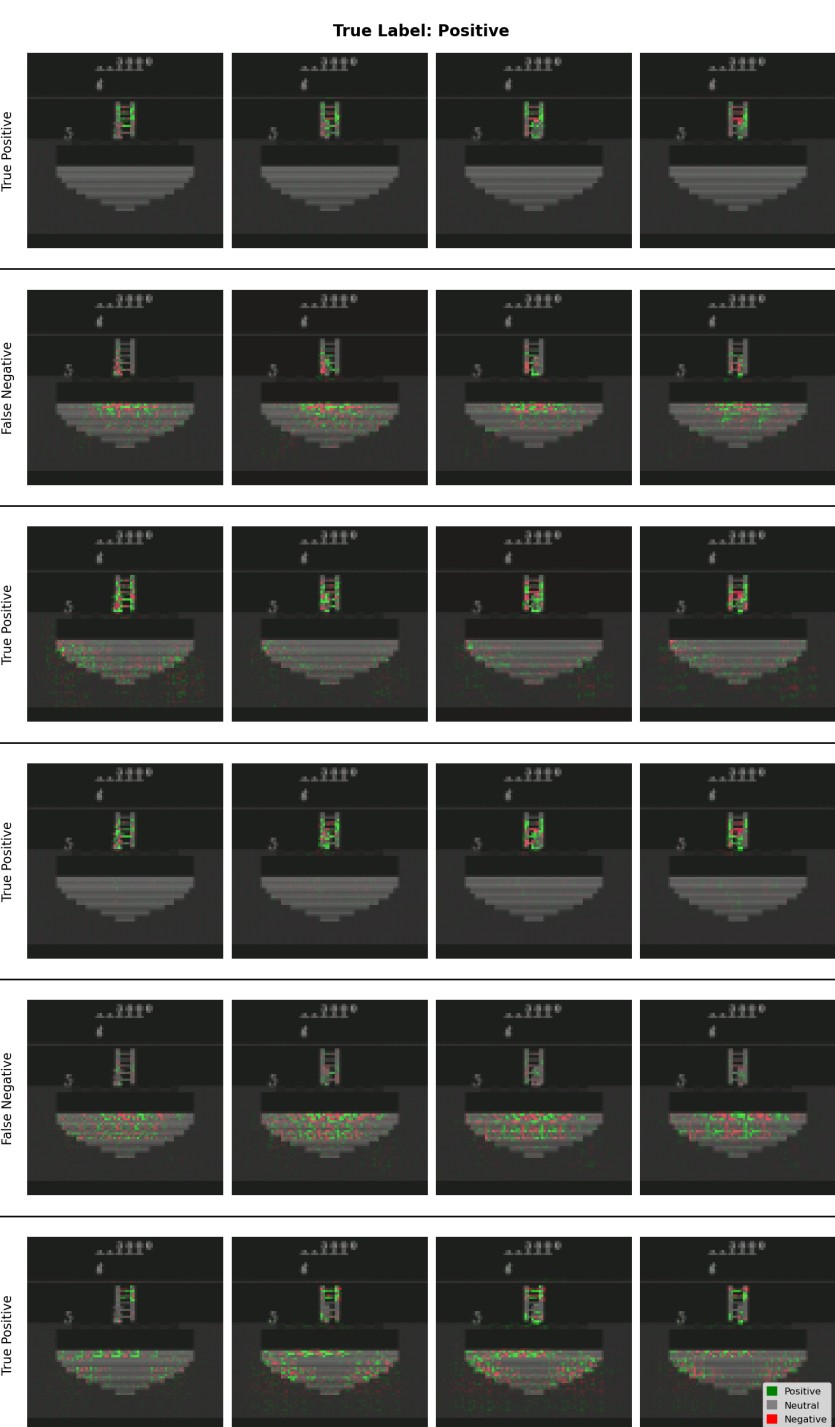

Figure 7: Gradient-based saliency maps for six ensemble members for the ClimbDownLadder subgoal. Green indicates features increasing predicted subgoal probability with red showing decreasing probability. Some members focus on features involving the ladder and agent while some focus on the lava below the floor. These distinct feature attributions confirm that ensemble members represent different hypotheses about causal structure, consistent with our method's goal of preserving multiple plausible interpretations from limited training data.

---

**Algorithm 2** MONTEZUMASREVENGE classifier experiment pseudocode

---

**Input:** room_datasets

  train_data ← [ ]
  test_performance ← [ ]
  **for** room_dataset in room_datasets **do**
     classifiers ← initialize new classifiers
     train_data.append(room_dataset)
     Train classifiers using train_data
     eval_performance ← [ ]
     **for each** room_dataset in room_datasets **do**
       room_eval ← classifiers accuracy on room_dataset
       eval_performance.append(room_eval)
     **end for**
     test_performance.append(max(ave(eval_performance)))        ▷ Record best classifier
  **end for**

---

---

**Algorithm 3** MONTEZUMASREVENGE policy experiment pseudocode

---

**Input:** room_datasets, max_steps_per_room, classifiers, rooms

  train_data ← [ ]
  test_performance ← [ ]
  **for** room_dataset in room_datasets **do**
     classifiers ← initialize new classifiers
     train_data.append(room_dataset)
     Train classifiers using train_data
     room_eval ← [ ]
     **for each** room in rooms **do**        ▷ Room is initiation state for policy training
       class_eval ← [ ]
       **for each** classifier in classifiers **do**
         policy ← initialize new policy
         steps ← 0
         **while** steps ¡ max_steps_per_room **do**
           steps_taken ← train policy for one episode
           steps ← steps + steps_taken
         **end while**
         success_rate ← [ ]
         **for** episode in 100 **do**
           *Get Manhattan distance between termination state from classifier and closest*
           *ground truth termination*
           Man_dist ← run policy for one episode
           success_rate.append(Man_dist)
         **end for**
         class_eval.append(ave(success_rate))
       **end for**
       room_eval.append(min(class_eval))        ▷ Record best member performance
     **end for**
     test_performance.append(ave(room_eval))
  **end for**

---

### C.3 CLASSIFIER SETUP

We use the `PyTorch` library for the classifier models. We use the `PyTorch` `nn.CrossEntropyLoss()` for our cross entropy loss and use the DivDis loss function from the original authors (available at `https://github.com/yoonholee/DivDis/tree/main`). There are more states outside the subgoal than inside so we use weight rescaling to balance weight updates. We do this using the `nn.CrossEntropyLoss() weights` parameter for this rebalancing. The Adam optimizer PyTorch implementation (`optim.Adam()`) and add L2 regulariza-

---

tion using the `weight_decay` parameter. We have included pseudocode for training the DivDis classifier in Algorithm 4. Training for the standard ensemble is the standard classifier training loop.

---

**Algorithm 4** Divdis classifier training pseudocode

---

**Input:** dataset, max_epochs, classifiers

  epoch $\leftarrow 0$
  **while** epoch<max_epochs **do**
    **for** batch in dataset **do**
      $x, u, y \leftarrow$ batch   $\triangleright x$ : labeled classifier input, $u$ : unlabeled classifier input, $y$ : true label
      unlabeled_pred $\leftarrow$ [ ]
      batch_labeled_loss $\leftarrow 0$
      **for each** classifier $f_i$ in classifiers **do**
        $\hat{y} \leftarrow$ classifier($x$)
        labeled_loss = labeled_loss + `nn.CrossEntropyLoss`($\hat{y}, y$)
        $\hat{u} \leftarrow$ classifier($u$)
        unlabeled_pred.append($\hat{u}$)
      **end for**
      divdis_loss $\leftarrow$ `DivDis_criterion`(unlabeled_pred)
      loss $\leftarrow$ batch_labeled_loss+divdis_loss
      `optimizer.step`(loss)               $\triangleright$ Update weights with respect to loss
    **end for**
    epoch $\leftarrow$ epoch+1
  **end while**

---

### C.3.1 MONTEZUMASREVENGE

The classifier architecture for each ensemble member and the single CNN is shown in Figure 8 (it is the same architecture for all models). The hyperparameters for MONTEZUMASREVENGE can be seen in Table 1. For Montezuma's Revenge, the state is a framestack of 4 timesteps and each frame is grayscale and resized to $84 \times 84$ as is consistent in the original Atari DQN experiments.

Labeled training data is collected by a human who moves the agent to different areas of each room in the MONTEZUMASREVENGE game for level 1. Because the data comprises of expert trajectories and the state consists of the previous four frames our labeled data set does not fully encompass the entire state space and it is very likely that a policy will encounter states that are not in this dataset during training. We use this labeled data as unlabeled data in our experiments, discarding the labels during training and use this data for evaluation during the classifier experiment. Note that while we evaluate and train on labeled data from all rooms that contain a ladder, we have data collected from rooms without a ladder so we can still provide the DivDis ensemble with unseen unlabeled data even when training on all ladder rooms.

### C.3.2 MINIGRID DOORMULTIKEY

The classifier architecture for each ensemble member and the single CNN is shown in Figure 9 (it is the same architecture for all models). The hyperparameters for MINIGRID DOORMULTIKEY can be seen in Table 2. The state is the fully observable, top-down RGB view of the grid resized to $84 \times 84$.

Labeled data collection is done in two ways for MINIGRID DOORMULTIKEY. First we move the agent to each accessible grid space (i.e. if the door is locked only grid spaces in the first room otherwise all grid spaces in both rooms), rotating to face each direction. The agent then collects the relevant key (e.g. if we are collecting data for COLLECTREDKEY we collect the red key) and again visits each accessible grid space. The agent unlocks the door and again visits each grid space. Data was also collected by randomly placing the agent and the available keys in different grid spaces as well as randomly setting the state of the door (open, unlocked and closed, locked and closed). We use labeled data as unlabeled data by discarding the labels during training.

## C.4 OPTION POLICY SETUP

We use the same DQN architecture for both MINIGRID and MONTEZUMASREVENGE, differing only in the number of actions. We use the Adam optimizer as implemented in PyTorch (`optim.Adam()`). Exploration is carried out using the `pfrl` library `LinearDecayEpsilonGreedy()`, a linearly decaying epsilon greedy explorer. We use the `pfrl` replay buffer implementation `PrioritizedReplayBuffer()` and model updates are carried out using the `ReplayUpdater()` also from `pfrl`. The DQN architecture is shown in Figure 10 and hyperparameters are displayed in Tables 3 and 4 for MONTEZUMASREVENGE and MINIGRID respectively. Our DQN model is implemented in PyTorch with a `pfrl` policy head. All experiments use $\gamma = 0.9$.

## C.5 HIGH-LEVEL POLICY SETUP

We use the PPO agent from the `pfrl` library. We use observation normalization, using `EmpiricalMormalization()` from `pfrl`. Optimization is done using the PyTorch `optim.Adam()` optimizer. The policy and value networks are shown in Figure 11, implemented in PyTorch with a `pfrl` policy head. Hyperparameters for DOORMULTIKEY MINIGRID are shown in Table 5. All experiments use $\gamma = 0.9$.

## C.6  MODEL ARCHITECTURES

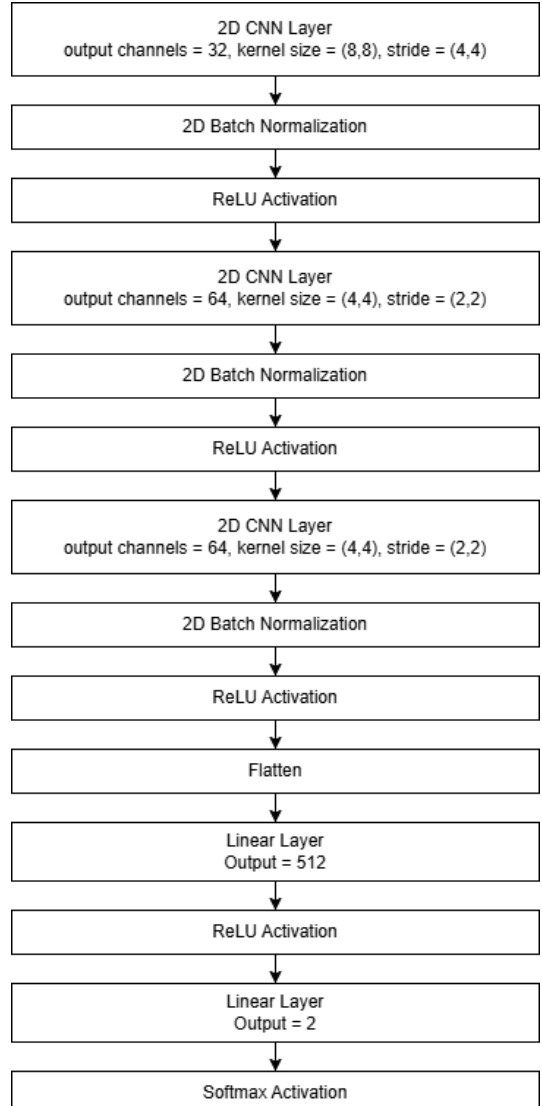

Figure 8: MONTEZUMASREVENGE classifier architecture.

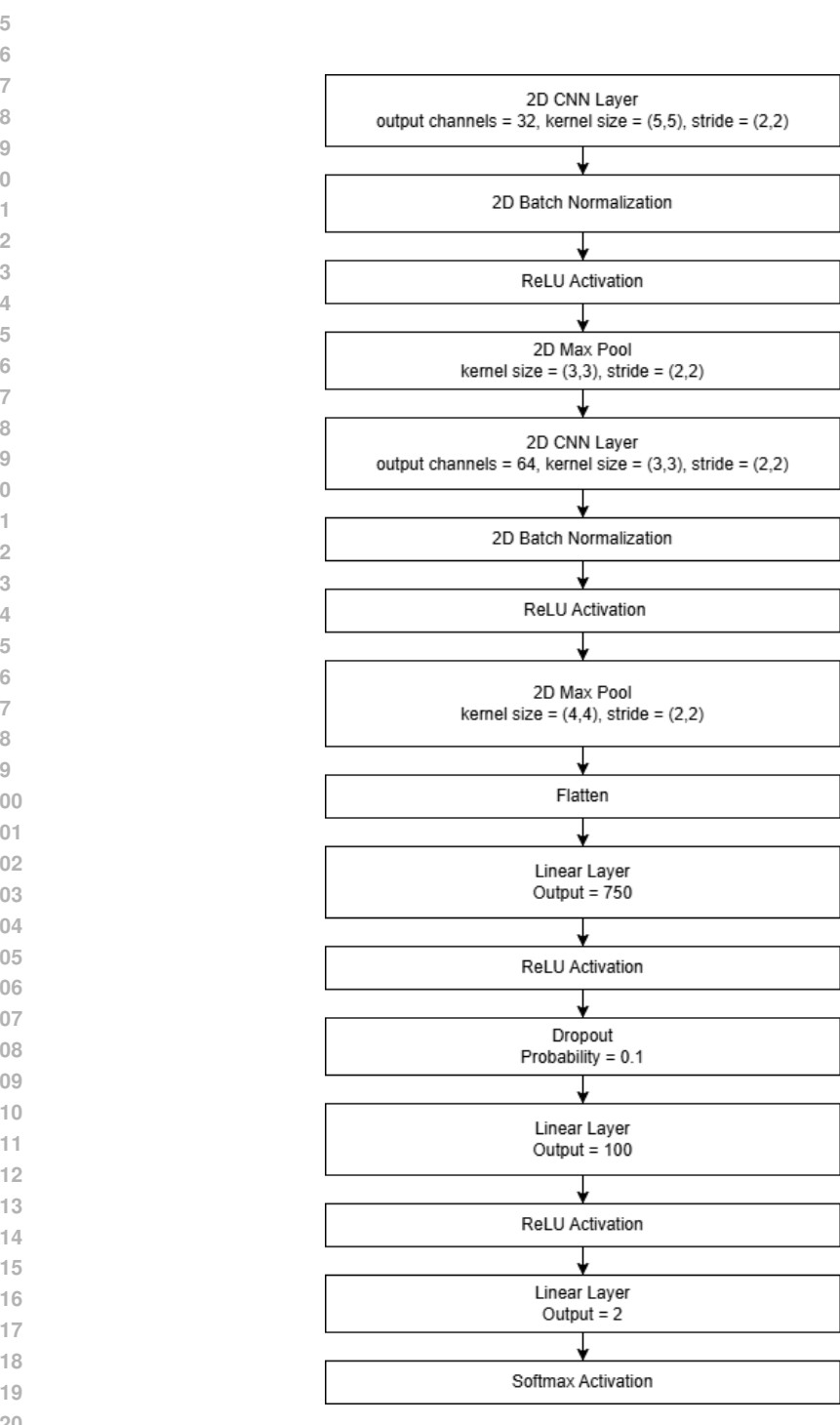

Figure 9: DOORMULTIKEY MINIGRID classifier architecture.

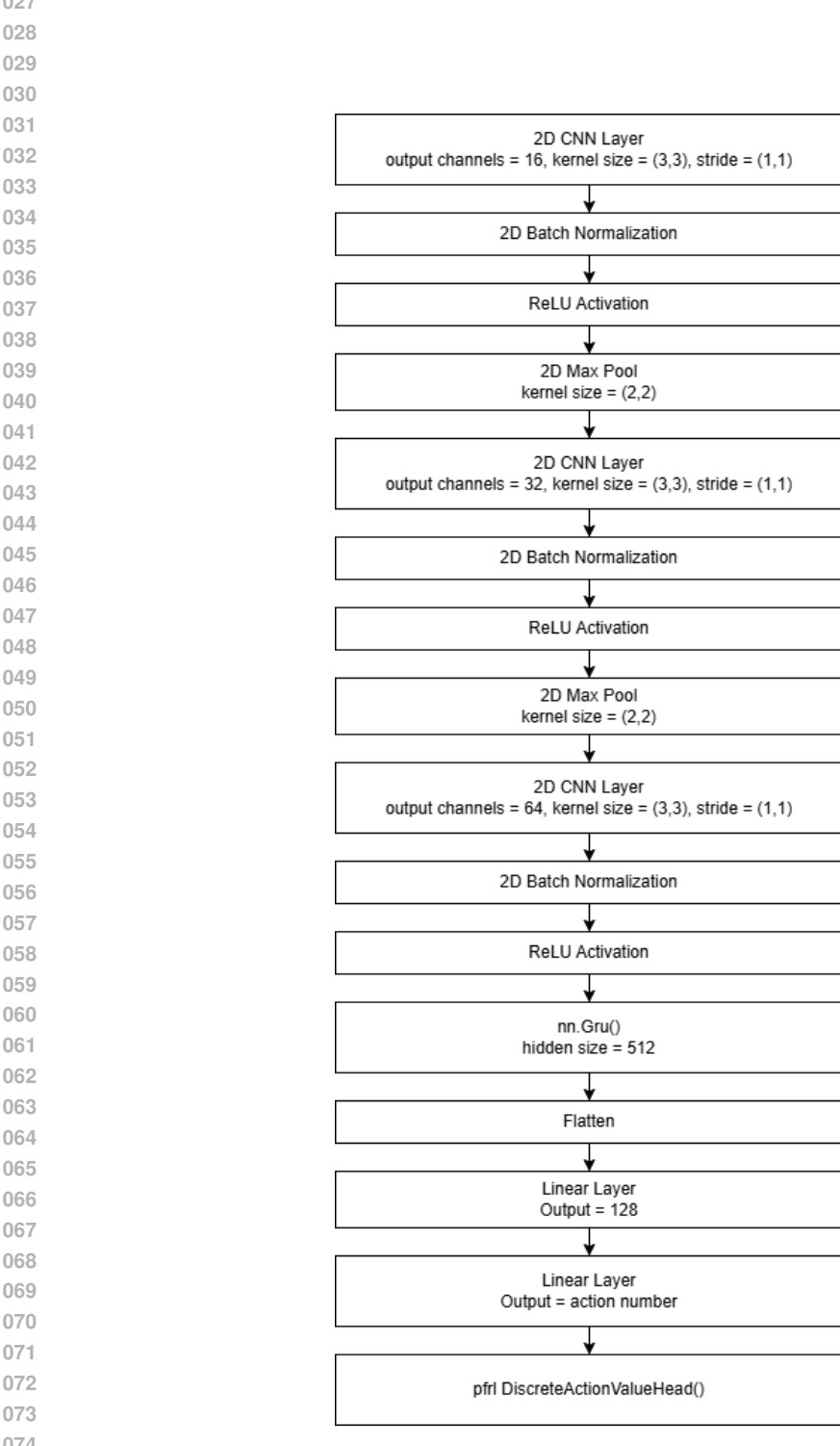

Figure 10: DQN architecture.

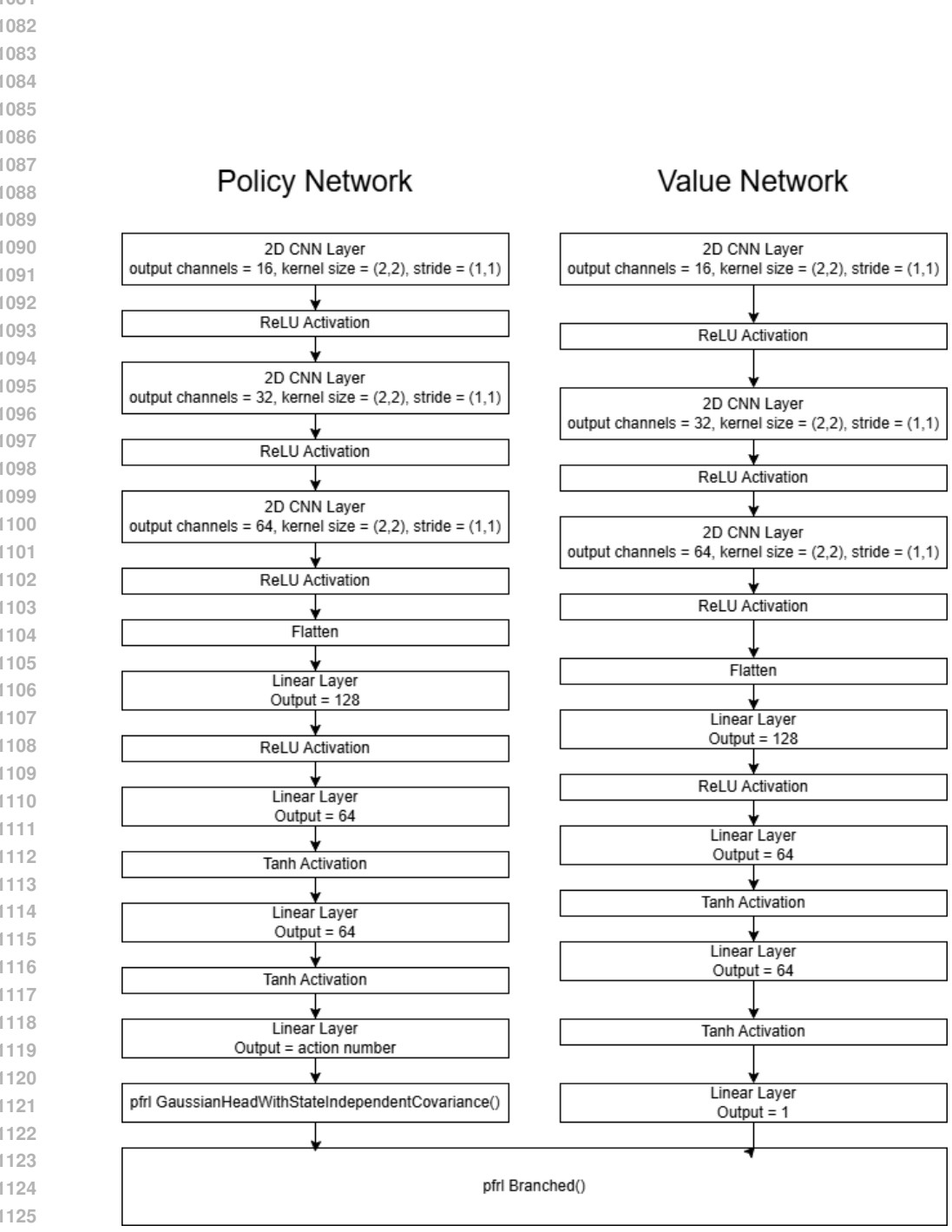

Figure 11: PPO architecture.

## C.7 Hyperparameters

Table 1: MONTEZUMASREVENGE classifier hyperparameters.

| Hyperparameter | DivDis | Standard | CNN |
|---|---|---|---|
| Learning Rate | $5 \times 10^{-4}$ | $5 \times 10^{-4}$ | $5 \times 10^{-4}$ |
| Diversity Weight | $3 \times 10^{-4}$ | 0.0 | $3 \times 10^{-4}$ |
| Ensemble Size | 6 | 6 | 1 |
| L2 Regularization Weight | $5 \times 10^{-4}$ | $5 \times 10^{-4}$ | $5 \times 10^{-4}$ |
| Batchsize | 64 | 64 | 64 |

Table 2: MINIGRID DOORMULTIKEY classifier hyperparameters.

| Hyperparameter | DivDis | Standard | CNN |
|---|---|---|---|
| Learning rate | $2 \times 10^{-4}$ | $2 \times 10^{-4}$ | $2 \times 10^{-4}$ |
| Diversity weight | $1 \times 10^{-4}$ | 0 | $1 \times 10^{-4}$ |
| Ensemble size | 3 | 3 | 1 |
| L2 regularization weight | $1 \times 10^{-4}$ | $1 \times 10^{-4}$ | $1 \times 10^{-4}$ |
| Batchsize | 64 | 64 | 64 |

Table 3: MONTEZUMASREVENGE DQN hyperparameters.

| Hyperparameter | Value |
|---|---|
| Replay buffer length | $1 \times 10^5$ |
| Update interval | 4 |
| Q-target update interval | 10 |
| Final Exploration frame | $4 \times 10^5$ decaying from 1 to 0.01 |
| Learning rate | $2.5 \times 10^{-4}$ |
| Batchsize | 32 |

Table 4: MINIGRID DOORMULTIKEY DQN Hyperparameters.

| Hyperparameter | Value |
|---|---|
| Replay buffer length | $1 \times 10^5$ |
| Update interval | 4 |
| Q-target update interval | 10 |
| Final Exploration frame | $8 \times 10^3$ decaying from 1 to 0.01 |
| Learning rate | $2.5 \times 10^{-4}$ |
| Batchsize | 32 |

Table 5: MINIGRID DOORMULTIKEY PPO Hyperparameters.

| Hyperparameter | Value |
|---|---|
| Replay buffer length | $1 \times 10^5$ |
| Update interval | 100 |
| Entropy coefficient | 0.01 |
| $\lambda$ | 0.97 |
| Batchsize | 64 |
| Epochs per update | 10 |
| Maximum L2 norm | 1 |
| Observation normalizer clip threshold | 5 |
| Standardize advantages | True |

